# Individual and Institutional Factors Contribute to Research Capacity Building for Early-Stage Investigators from Groups Underrepresented in Biomedical Research: A Qualitative Comparative Analysis

**DOI:** 10.3390/ijerph20095662

**Published:** 2023-04-27

**Authors:** Yulia A. Levites Strekalova, Diana L. Kornetti, Ruixuan Wang, Adriana Báez, Lee S. Caplan, Muhammed Y. Idris, Kimberly Lawson, Jada Holmes, Mohamed Mubasher, Priscilla Pemu, Jonathan K. Stiles, Maritza Salazar Campo, Alexander Quarshie, Thomas Pearson, Elizabeth O. Ofili

**Affiliations:** 1Clinical Translational Science Institute, University of Florida College of Public Health and Health Professions, Gainesville, FL 32610, USA; 2Departments of Pharmacology and Otolaryngology, School of Medicine, University of Puerto Rico, San Juan 00936, Puerto Rico; 3Department of Medicine, Microbiology, Biochemistry and Immunology, Clinical Research Center, Morehouse School of Medicine, Atlanta, GA 30310, USA; 4Department of Organization and Management, University of California, Irvine, CA 92093, USA

**Keywords:** grant writing coaching, diversity in biomedical research, early-stage investigators, research capacity building

## Abstract

Background: Enhancement of diversity within the U.S. research workforce is a recognized need and priority at a national level. Existing comprehensive programs, such as the National Research Mentoring Network (NRMN) and Research Centers in Minority Institutions (RCMI), have the dual focus of building institutional research capacity and promoting investigator self-efficacy through mentoring and training. Methods: A qualitative comparative analysis was used to identify the combination of factors that explain the success and failure to submit a grant proposal by investigators underrepresented in biomedical research from the RCMI and non-RCMI institutions. The records of 211 participants enrolled in the NRMN Strategic Empowerment Tailored for Health Equity Investigators (NRMN-SETH) program were reviewed, and data for 79 early-stage, underrepresented faculty investigators from RCMI (n = 23) and non-RCMI (n = 56) institutions were included. Results: Institutional membership (RCMI vs. non-RCMI) was used as a possible predictive factor and emerged as a contributing factor for all of the analyses. Access to local mentors was predictive of a successful grant submission for RCMI investigators, while underrepresented investigators at non-RCMI institutions who succeeded with submitting grants still lacked access to local mentors. Conclusion: Institutional contexts contribute to the grant writing experiences of investigators underrepresented in biomedical research.

## 1. Introduction

In order to address the current health disparities and improve minority health outcomes so as to improve overall population health, increased effectiveness of the research workforce is essential [1,2]. Enhancement of diversity within the U.S. research workforce is a recognized need and priority at a national level. With the increasing racial diversity of the U.S. population, an unequivocal underrepresentation of minority scientists and faculty participating exists in biomedical research [1,3,4,5,6]. Existing research shows that a diversity of perspectives is directly associated with innovation and impact of research, but several groups, such as investigators from racial and ethnic groups, individuals from economically disadvantaged backgrounds, people with disabilities, and women, remain underrepresented in research due to individual and institutional challenges.

Focusing on the diversity of the biomedical research workforce and taking a comprehensive systemic approach [7], several agencies including the National Institutes of Health (NIH) have sponsored strategic initiatives in support of increasing the number of NIH-funded investigators from the groups currently underrepresented in research [8], and increase and diversify the biomedical research workforce. The most recent efforts included programs around minority recruitment and the development of junior faculty [1], mentoring [2,3,9,10,11,12], grant writing training [4,6,13,14], and coaching [5,15]. Two of the large NIH-funded programs, the National Research Mentoring Network (NRMN) and the Research Centers in Minority Institutions (RCMI), focus on institutional research capacity building, as well as the mentoring and training of early-stage investigators (ESI) in support of junior faculty success and progression to tenure [5,6,11,14]. In the following paragraphs, we provide a brief overview of these programs and formulate the research question to investigate the interrelation of individual and institutional factors that affect the progress of submitting NIH grant applications by ESIs from groups underrepresented in biomedical research.

### 1.1. Research Centers in Minority Institutions

The RCMI program was established by the U.S. Congress in 1985 to serve an integral role in the accelerated representation of underrepresented groups in biomedical and health disparity research. The program is currently administered by the National Institute on Minority Health and Health Disparities (NIMHD) and is tasked with strengthening research training, conduct, and infrastructure in minority-serving colleges and universities with the goal of developing a cadre of independent investigators from groups traditionally underrepresented in research [16,17,18,19,20]. Currently, there are 22 RCMI institutions and a Coordinating Center funded through the NIMHD. Each RCMI is structured with four distinct cores: administrative core (AC), community engagement core (CEC), investigator development core (IDC), and research infrastructure core (RIC). RCMI-funded work must address: (1) enhanced capacity in basic biomedical, behavioral, and/or clinical research; (2) training and support of affiliated investigators to become more successful in obtaining extramural funding, especially from the NIH, to address diseases disproportionately impacting target populations; (3) developing new and early career investigators; (4) enhanced quality of research on minority health and health disparities; and (5) establishing and sustaining relationships with community-based partners [17]. By preparing future generations of scientists at institutions with historical contributions from minority scholars in medicine, science, and technology, the RCMI program seeks to accelerate the delivery of solutions that address health disparities in those communities that are most impacted [21]. RCMI supports the vision of NIMHD to advance the science of minority health and health disparities research through rigorous, mentored research experiences for all program investigators [17]. In a report titled “Rising Above the Gathering Storm: Energizing and Employing America for a Brighter Economic Future”, the National Academies (2005) stated that a strong, talented, and innovative science and technology workforce is essential for the United States to maintain global leadership and competitiveness. Investment in minority students and those institutions that train and prepare them will have direct implications on the U.S. economy, security, and prosperity in years to come. RCMI was developed to address workforce disparity, improve the quality of scientific inquiry, and promote collaborative research focused on improving minority health and reducing long-standing health disparities [20]. Although funding from the RCMI program has made previously resource-limited institutions and schools of health professions an equal partner with those research-intensive majority institutions to acquire the knowledge necessary for the improvement of national health in the U.S., even today, some 17 years after this presentation to Congress, workforce diversity remains below that represented in the general population, lagging in its contributions to improving health status and equity of care [1,20,22], with continued discrepancies in research grant success rates between White and Black applicants [13,21].

RCMI efforts specific to investigator development aim to create a supportive career development environment for postdoctoral fellows, junior faculty, and other ESIs to conduct research, with particular support for women, people with disabilities, and those from underrepresented racial and ethnic groups [20]. Furthermore, the RCMI program operates under the mandate of building the institutional capacity for research while facilitating meaningful and sustained engagement with community stakeholders representing racial and ethnic minorities and other populations that disproportionately experience adverse health outcomes [17,20]. With an explicit focus on health equity research, RCMI community engagement efforts include: (1) supporting community-engaged research, (2) translating and disseminating findings, (3) developing partnerships, and (4) building capacity around community research [17]. To achieve these goals, it is essential that trust is established between community and academic collaborators, and ongoing mutually beneficial relationships are developed. Akintobi et al. (2021) reported that successful community–academic partnerships are dependent on building trust and rapport within minority communities to combat historically negative experiences and failure to protect research participants that have occurred.

RCMI capacity-building programs provide an opportunity for the improved coordination and leveraging of resources across all RCMI and interinstitutional collaborative partnerships [18]. However, key barriers have been reported by RCMI investigator development leads, which include a limited number of NIH-funded senior faculty to serve as role models and mentors, junior faculty lacking knowledge about finding funding and training opportunities, lack of pre-award grant application preparation and internal review systems, and lack of bridging funds until funding decisions are made [20]. Challenges exist in the development of community-driven and culturally relevant solutions for addressing health inequities, as well as bolstering up junior faculty and early investigators working within research institutions and their interinstitutional collaborative partners. In addition, there are barriers at the institutional level. These include hiring freezes, limited start-up funds for newly recruited faculty, leadership turnover, insufficient grant development support, and heavy teaching and service loads [20]. Finally, Ofili et al. (2021) reported that predominantly white or majority institutional cultures may lack the necessary elements of inclusion and equity to consistently send the message that racial and ethnic minorities, women, and people with disabilities belong in science. To better understand and share effective strategies among RCMIs, improved evaluation tools, metrics, and methodologies are needed. Further research investigating the interrelationships of individual and institutional factors that affect the research capacity of RCMI investigators may provide essential evidence for future program planning and implementation [16,17,19,20,21].

### 1.2. National Research Mentoring Network

NRMN is a more recent NIH initiative that was established in 2014 as a trans institutional project, funded by the NIH Common Fund and managed by the National Institute of General Medical Sciences (NIGMS). NRMN seeks to increase access to mentoring across career stages, improve mentoring relationships through training, increase access to resources and career development, and promote the value of mentoring to the science community [2,5,15]. Specific attention has been paid to mentoring processes, as the research literature has documented their benefits, including increased research productivity, faster academic promotion, and career satisfaction [2,9]. Inadequate mentoring is a contributing factor in the underrepresentation of minorities in biomedical research, driving the development of MyNRMN, a social platform designed to facilitate the connection between mentors and mentees [2,6]. Studies have supported the importance of multiple mentors in the success of young researchers (mentees) [3]. Ahmed et al. (2020) described and assessed the functionality of MyNRMN to support the formation of mentoring networks, including the development of connections among peer mentors, and found that the platform allowed mentees flexibility and options to engage with individuals in different areas of expertise and specializations. In mentoring, membership in social networks such as MyNRMN can serve as a measure of mentee social capital and provide access to enriched mentoring experiences through an extended network of connections [3]. Additionally, partnerships with minority-serving institutions (MSIs) and academic medical centers are of value in the mentoring process. Hemming et al. (2019) reported that an institutional investment in research mentoring is critical, stating that resource scarcity, adequate training, appropriate time allotments, and formal recognition are key components when establishing an institutional policy and mentoring program. Choi et al. (2019) stated that a culture of mentorship should be a major strategic priority for academic medical centers, with a declining rate of physician-scientists in the workforce due to a lack of mentor availability. Mentoring is a key strategy for developing the next generation of faculty leaders and promotes institutional sustainability through increasing research productivity and external funding opportunities [9].

In addition to mentoring, ESIs require coaching to model successful grant-writing practices. Although many grant writing programs exist, because of the competitive nature of this process, their design and preparation require extensive time and effort that frequently extend well beyond the parameters of any such program [4,5,13,14,15]. Of greater concern for the biomedical research workforce are the well-documented disparities in the rates of NIH proposals submitted by, and grants awarded to, members of underrepresented groups [6,13]. Specifically, Thorpe et al. (2020) reported that African American investigators submitted fewer initial R01 grant applications, received lower overall priority scores, and resubmitted unfunded grants less frequently compared with their White peers, in part due to a lower institutional capacity to provide research support and mentoring. Furthermore, there is a need for sustained, long-term coaching and mentoring pre-submission support for the review of proposal drafts (e.g., complete proposal reviews or mock review sessions) and post-submission activities (e.g., coaching on resubmission introductions and revisions) [4,6,13,14].

### 1.3. Research Questions

In our study, we investigated the results of participation in grant writing coaching by investigators underrepresented in research from RCMI and non-RCMI institutions to answer the following research questions:

RQ1: What combinations of individual and institutional factors predict the successful NIH grant application by investigators underrepresented in research?

RQ2: What combinations of individual and institutional factors predict the failure to submit an NIH grant application by investigators underrepresented in research?

RQ3: To what extent do the individual and institutional factors that affect grant writing outcomes differ for investigators from RCMI and non-RCMI institutions?

## 2. Materials and Methods

A qualitative comparative analysis was used to address the research questions of this study.

### 2.1. Overview of Qualitative Comparative Analysis

Briefly, qualitative comparative analysis (QCA) is a mixed-method approach that investigates qualitative and/or quantitative data for a set of cases to identify the combination of factors that explain the presence or absence of an outcome of interest [23,24]. QCA is most appropriate to generate hypotheses and explore research questions that focus on complex social phenomena [25,26]. QCA uses set theory to identify non-statistical relationships among the included cases. The approach is different from qualitative methods in that it provides a formal and systematic method to identify a set of factors and conduct a cross-case comparison. QCA is also distinct from statistical methods in that it focuses on the combination of factors rather than on the explanatory power of individual variables. In fact, the methodological guidelines for QCA explicitly recommend the use of the term condition to distinguish it from the term variable, which is used in statistics [23].

QCA is guided by three assumptions. First, the *equifinality* assumption recognizes that more than one combination of factors may lead to the investigated outcome [26]. To illustrate, multiple experiences can lead to an appointment to a faculty position, including varying years of postdoctoral training, number of publications, and teaching experiences. Combinations of these factors can shape different pathways to a faculty appointment, which is representative of equifinality. Conversely, a *unifinal* focus would result in identifying the average years of training, average number of publications, and average number of courses taught to describe the factors predictive of appointment to a faculty position. Second, *conjunctural causation* assumes that an individual factor may not be sufficient to produce an outcome, but that it needs to be investigated in combination with other factors [26]. In the faculty appointment example, postdoctoral training, publications, and teaching need to be evaluated holistically and in combination rather than being investigated for their independent effects. Finally, asymmetrical causation assumes that the occurrence and non-occurrence of an outcome are distinct phenomena, and for a set of factors that predict an outcome of interest, one cannot assume that the absence of these factors predict a non-occurrence of the outcome [26]. For example, if we find that the prior teaching record predicts an appointment to a faculty position, we cannot automatically assume that a lack of teaching experience will result in nonappointment.

### 2.2. Study Participants

Data for this study were collected from applications and surveys completed by participants in the NRMN Strategic Empowerment Tailored for Health Equity Investigators (NRMN-SETH). We have previously reported on the overall design of the NRMN-SETH program. Briefly, this program was offered to five cohorts of early-stage and new investigators to support them in developing and submitting an independent NIH grant application. Participation for the last cohort had not concluded at the time of writing this manuscript; therefore, application and outcome data for cohorts 1–4 were included in the analyses. In total, the records of 211 participants enrolled in grant writing cohorts 1–4 were reviewed for inclusion in the analysis. The inclusion criteria were (1) position of an assistant professor (or equivalent), and (2) belonging to a group underrepresented in biomedical research (operationally defied as identifying as Black/African American, Latino(a)/Hispanic, American Indian/Alaska Native, Native Hawaiian, or Pacific Islander OR reporting a disability status OR identifying as a first-generation college graduate), which resulted in the inclusion of 79 early-stage, underrepresented faculty investigators from RCMI (n = 23) and non-RCMI (n = 56) institutions. Summary information about the participants is presented in Table 1.

### 2.3. Explanatory Factors and Outcome of Interest

The inclusion of explanatory factors in the QCA analysis is guided by prior research and theory. For our study, we were guided by the logic model of the NRMN-SETH program, which that considers the combination of personal characteristics, such as demographics, academic position, and self-efficacy of research, as well as research environment, such as institutional support and access to mentors. The following factors were included in the analyses: RCMI or non-RCMI home institution (RCMI); perceived high or low level of institutional support (support); availability of access to a mentor at home institution (mentor); availability for high (over 16 h/week) or low (15 h/week and less) time commitment to grant writing, determined from the population mean (effort); high (15 and more papers) or low (under 15 papers) research record at the time of the NRMN program participation, determined as the number of publications below or above the NRMN-SETH population mean (record); and high (four or more courses per year) or low (three or fewer courses per year) (teaching). We looked at the submission of an NIH grant as the outcome of interest (submit). All data were converted and presented as a truth table with conditions coded as 1 or 0 for crisp-set QCA [27,28].

## 3. Results

The results of the analyses presented below use a standard QCA notation, where capital letters (e.g., RCMI) indicate that the presence of a factor (or value of 1) explains the outcome, and small letters (e.g., rcmi) indicate that the absence of a factor (or value of 0) explains the outcome. The combination of contributing factors (logical operand AND) is represented by an asterisk, ‘*’.

### 3.1. Factors Contributing to Successful Grant Submissions

RQ1 focused on identifying the combinations of factors that contribute to successful grant submissions by investigators from RCMI and non-RCMI institutions that are underrepresented in biomedical research. Five combinations of factors (Figure 1, where * represents the logical operand AND), two for RCMI institutions and three for non-RCMI institutions, were identified as explaining a successful grant submission. For RCMI institutions, the record of previous publications and access to mentoring were factors common to both solutions. Interestingly, perceived institutional support and teaching assignments emerged as counterfactors, in that investigators with high teaching assignments who have submitted grants perceived a lower level of institutional support and vice versa. Finally, low committed effort appeared as an explanatory factor for investigators with a high teaching effort, but did not appear as an explanatory factor for investigators with a low teaching effort.

For investigators from non-RCMI institutions (rcmi), institutional support, access to mentoring, committed grant writing effort, and research record were identified as consistent factors explaining the success in submitting grant applications. Interestingly, the lack of mentoring, rather than its availability, was the explanatory factor. Committed effort and research record emerged as counterfactors, and, similar to the RCMI investigators, a high teaching load was associated with low perceived institutional support.

### 3.2. Factors Contributing to a Failure to Submit a Grant Application

RQ2 focused on identifying the combinations of factors that explain the failure to submit a grant application. Similar to RQ1, home institution RCMI or non-RCMI was a contributing factor, which indicates that the experiences of investigators at these two types of institutions varied (see Figure 2 for the combination of factors). For RCMI institutions, the low publication record and the high teaching load were predictive of the inability to submit an NIH grant application. Interestingly, high perceived institutional support also appeared as a contributing factor. This indicates that institutional support alone does not predict grant submissions, but should be considered in combination with other individual factors. For investigators from non-RCMI institutions, one combination of factors was identified. Not surprisingly, low perceived support, lack of local mentoring, lack of research record, and high teaching load contributed to a failure to submit a grant application.

### 3.3. Differences among Investigators from RCMI and non-RCMI Institutions

RQ3 focused on evaluating the differences in factors that affect the success of faculty in submitting grant applications, differing for RCMI and non-RCMI investigators. The analyses for RQ1 and RQ2 indicated that the home institution is a contributing condition for explaining the experiences in grant writing for underrepresented investigators. The analyses also indicated that access to a local mentor is a predictor of success for RCMI investigators, while investigators at non-RCMI institutions were able to succeed without access to a local mentor. However, the analyses also indicated that the research record and teaching load are conjunctural factors that are common to investigators in both types of academic institutions. Figure 3 presents the results visually.

## 4. Discussion

This study explored the factors related to the success or failure to submit a grant during participation in a structured grant writing coaching program through the NRMN. In particular, this study explored the experiences of investigators underrepresented in biomedical research and contributed to the science of biomedical workforce diversity. Recognizing the complexity and multiplicity of the pathways that result in a grant being submitted, this study used a qualitative comparative analysis approach to systematically investigate the configurations of individual and institutional conditions. It is important to acknowledge that QCA is a set-theoretic method, and the results of the analyses depended on the characteristics of the included cases. In addition, QCA is not a statistical approach, and the results do not show the relative strength of the identified predictors. Rather, the results provide a systematic assessment of the complexity of grant writing capacity building for underrepresented investigators based on the currently available evidence and provide a foundation for generating hypotheses for future investigations and interventions. Additionally, QCA is a mixed-method approach that aims to reduce qualitative data to a set of variables for algebraic analyses. As such, this method does not include thematic or qualitative content analyses of comments or transcripts.

This study provided several insights. First, institutional membership (RCMI vs. non-RCMI) emerged as a persistent predictive factor in all outcome configurations, signaling that home institution as a condition carries an explanatory role in explaining the differences in ESI investigator experiences. Combinations in which the home institution condition appeared in the analyses provide additional insights into the pathways to grant submissions for investigators underrepresented in biomedical research. Most notably, the analyses showed that for the investigators at the RCMI institutions, access to a local mentor contributed to the success in submitting a grant application, while investigators at non-RCMI institutions needed to overcome the challenge of not having access to a local research mentor.

The objective of this study was to understand the combination of factors that affect the success of ESIs in submitting a grant proposal. It is also essential to recognize that ESIs in biomedical sciences at all research institutions experience a variety of pressures from institutional and departmental leaders to submit grant proposals. Institutions and departments often have specific funding goals that they want to achieve, and they may encourage or even require ESIs to submit grant proposals to help meet those goals. ESIs are often expected to produce research output at a high level, and grant funding is one of the key metrics through which research productivity is measured. Institutions and departments may pressure ESIs to submit grant proposals to ensure that they meet research expectations. It is not uncommon for academic institutions to state the goals for faculty research productivity in expected level of funding and effort coverage, and these expectations can also be linked directly to faculty performance evaluations and progress toward tenure and promotion. Therefore, grant funding becomes a key metric for ESIs’ career advancement, as it can lead to or stall promotions, tenure, and other professional opportunities. This push for quantitative markers of productivity may also create pressure for ESIs to submit proposals even if they are not fully prepared or have not fully developed their research ideas.

Practically, it is important for institutions and departments to develop structures that support ESIs in their grant proposal submissions, while also recognizing the pressures they face. ESIs should have adequate time and resources to prepare their proposals, and institutional and departmental leaders should work to create a supportive environment that encourages risk taking and innovation in research. Additionally, mentoring and training programs should be in place to help ESIs develop the skills necessary to successfully navigate the grant application process. The findings of this study support the following recommendations for developing the nurturing professional development environment for ESI faculty from groups underrepresented in biomedical sciences. First, the department should assign committed research mentors to the ESI faculty rather than leaving the mentor search to the faculty. The lack of mentors will signal to the institutions that additional investments are necessary in the development of the existing faculty. This can often be achieved by providing protected time to senior faculty to signal the recognition and expectation of their role as mentors. Second, this study showed that pre-grant writing publication productivity was a predictor of a grant submission. Therefore, institutions can set up academic writing programs that provide scaffolding for ESIs to publish papers, develop institutional networks, and become acquainted with mentors. Together, these two interventions will likely address the perception of low institutional support, which, as this study showed, also contributes to failed attempts to submit a grant.

This study provides fundamental interinstitutional evidence on the individual and institutional factors that affect the rates of grant writing among ESIs, and it has also identified the need to explore additional questions. Future studies are needed to explore the differences in access to mentoring, perceived institutional support, and research cultures by underrepresented investigators at minority-serving and predominantly White institutions. Furthermore, focused investigation on the researcher identity and institutional culture can provide additional understanding about the mentoring networks available to researchers underrepresented in biomedical research. Additional studies are needed to assess the contributions of external developers and mentoring networks to the success of underrepresented investigators. This may also indicate the need for changes or additional considerations in the review criteria of NIH career grants. Currently, the presence of local mentors is a score-generating criterion, whereas the absence of local mentors and lower scores may contribute to lower funding rates for underrepresented investigators. At the institutional level, studies that compare the factors that support grant writing among established and new investigators can provide an understanding of other structural and cultural factors that ultimately contribute to the success of diverse cohorts of biomedical investigators.

## 5. Conclusions

This article adds to the growing body of literature on the science of biomedical workforce diversity. The achievement of independent grant funding is one of the basic metrics of success for biomedical researchers, but the pathways to this outcome are influenced by a variety of institutional and individual conditions. The qualitative comparative analysis reported in this article provides strong evidence that the research infrastructure created by the RCMI network makes a difference in researcher experiences by providing access to local mentors. Continued investment in and support for the RCMI network is necessary to reduce isolation and diversify the biomedical research workforce.

## Figures and Tables

**Figure 1 ijerph-20-05662-f001:**
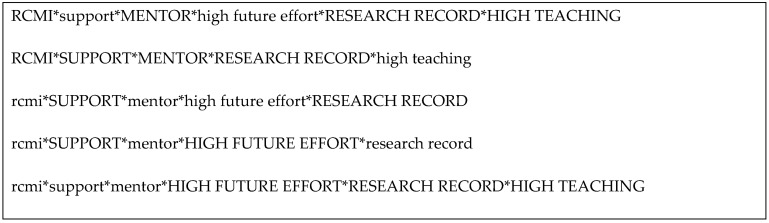
Factors contributing to successful grant submissions.

**Figure 2 ijerph-20-05662-f002:**
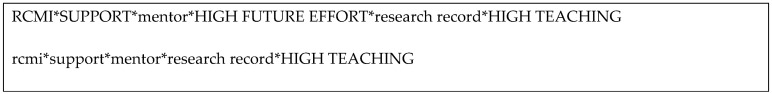
Factors contributing to the failure to submit a grant application.

**Figure 3 ijerph-20-05662-f003:**
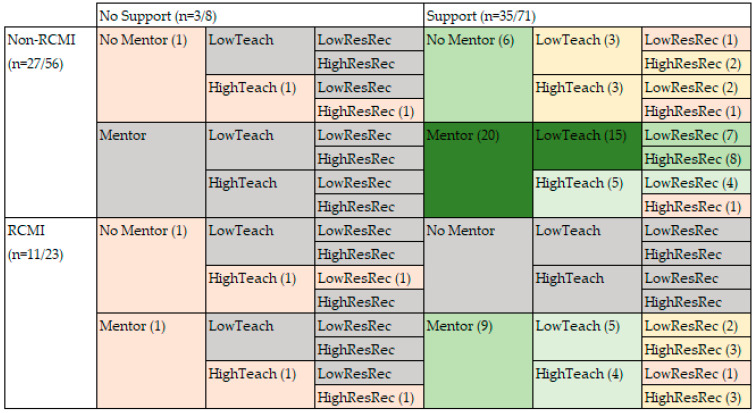
Diagram of factors contributing to grant submission. Numbers in parentheses indicating the number of participants for each combination of factors, and colors indicating no (gray), low (red), moderate (yellow), and high (green), and very high (dark green) recurrence.

**Table 1 ijerph-20-05662-t001:** Demographic and organizational characteristics of the study participants.

	Cohort				
	1	2	3	4	TOTAL
Participants at the RCMI Institution					
No	22	2	7	25	56
Yes	5	6	7	5	23
TOTAL FOR COHORT	27	8	14	30	79
Participant Race/Ethnicity					
African American (Black)	18	4	11	15	48
Asian				2	2
Hispanic or Latino/Latina	9	3	3	6	21
Native American				1	1
More than one race		1		1	2
White				5	5
Participant Gender					
Female		6	10	24	40
Male		2	4	5	11
Other				1	1
Data not collected	27				27
Participant First Generation Status					
No				13	13
Yes				17	17
Data not collected	27	8	14		49
Participants Identifying as Individuals with a Disability					
No				26	26
Yes				4	4
Data not collected	27	8	14		49
Participants with an Identified Mentor					
No	7		2	11	20
Yes	20	8	12	19	59
Participants with a High Teaching Assignment (two or more courses each semester)					
No	15	3	7	20	45
Yes	12	5	7	10	34

## Data Availability

The data presented in this study are available on request from the corresponding author. The data are not publicly available due to privacy concerns.

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
