# Peer review of "Individual and Institutional Factors Contribute to Research Capacity Building for Early-Stage Investigators from Groups Underrepresented in Biomedical Research: A Qualitative Comparative Analysis"

_ijerph, 2023, doi:10.3390/ijerph20095662_

Round 1

Reviewer 1 Report

The authors focused on the success and failure rate of the NIH grants using some early stage researcher. Authors neglected some important issues related to this study such as

The authors should present some graphical for this research

Authors should add the comparative data based on applicant age to show the difference between experienced researchers NIH grant winning rate and early stage researchers

The authors have to discuss the university/leader stress on early stage researchers for these grants and future prospective for them

Author Response

We thank the reviewers for their thorough and generous feedback. Below are our responses to their comments.

Response to Reviewers

Reviewer 1

The authors focused on the success and failure rate of the NIH grants using some early stage researcher. Authors neglected some important issues related to this study such as

The authors should present some graphical for this research

Response: Figure 3 has been added to provide visual summary of the results.

Authors should add the comparative data based on applicant age to show the difference between experienced researchers NIH grant winning rate and early stage researchers

Response: the focus of this paper was the experiences of ESI investigators in RCMI and non-RCMI institutions. Furthermore, the NRMN program targets early-stage and new NIH investigators. We have added the suggestion for longitudinal tracking of outcomes to the discussion.

The authors have to discuss the university/leader stress on early stage researchers for these grants and future prospective for them

Response: we have added this topic as a possible future research agenda.

Reviewer 2 Report

Please view the attachment.

Author Response

Reviewer 2

This paper presents a qualitative study of the factors that play important roles in the research capacity building of early-stage investigators from underrepresented groups in biomedical research.

The overall paper content is well-written with sufficient background information about the study.

The study methods and results can be improved with better visualization.

Response: this comment echoes the suggestion from Reviewer 1. Figure 3 has been added to the paper.

The first and second research question seems to complement each other. When addressing these two research questions, it would be helpful to have a diagram of the factors being considered and their relative combination that leads to the answer to these research questions.

Response: We clarified the presentation of the results. We also feel that Figure 3 addresses this comment too.

To gain a better understanding of the participants involved in this study, visualization of the demographic and relevant background information can be presented.

Response: a summary demographics table has been added.

For the QCA analysis, a summary table of the collected data and the analysis process should be presented either in the main text or the supplementary information for readers to better understand the process.

Response: we have expended the methods and results sections to address this comment.

Figures in the Results section is currently full of text instead of a proper visualization, which should also be improved. As a qualitative study, analysis of interview response from the participants can also be helpful to gain insights from their perspective.

Response: summary table and illustrative figure have been added.

Directly quoting some of selected representative responses followed by comments related to the research questions can be helpful. It would be better to provide a list of actionable items based on the study result to improve the status quo.

Response: We have expended the discussion to address this comment. 

A discussion on the potential application of this study into other relevant fields can also be helpful.

Response: We have expended the discussion to address this comment. 

In summary, the work in this paper is interesting but the presentation can be further improved.

Response: We thank the reviewer for specific and useful feedback!

Round 2

Reviewer 1 Report

Great

Reviewer 2 Report

Thank you for addressing the comments. The manuscript has been significantly improved.